# Efficient excitation and control of integrated photonic circuits with virtual critical coupling

Jakob Hinney[1,5], Seunghwi Kim [2,5], Graydon J. K. Flatt [3], Ipshita Datta[1], Andrea Alù [2,4] ✉ & Michal Lipson [1,3] ✉

Critical coupling in integrated photonic devices enables the efficient transfer of energy from a waveguide to a resonator, a key operation for many applications. This condition is achieved when the resonator loss rate is equal to the coupling rate to the bus waveguide. Carefully matching these quantities is challenging in practice, due to variations in the resonator properties resulting from fabrication and external conditions. Here, we demonstrate that efficient energy transfer to a non-critically coupled resonator can be achieved by tailoring the excitation signal in time. We rely on excitations oscillating at complex frequencies to load an otherwise overcoupled resonator, demonstrating that a virtual critical coupling condition is achieved if the imaginary part of the complex frequency equals the mismatch between loss and coupling rate. We probe a microring resonator with tailored pulses and observe a minimum intensity transmission $T = 0.11$ in contrast to a continuous-wave transmission $T = 0.58$, corresponding to 8 times enhancement of intracavity intensity. Our technique opens opportunities for enhancing and controlling on-demand light-matter interactions for linear and nonlinear photonic platforms.

Efficient coupling between waveguides and resonators is of critical importance in integrated photonic systems, but its implementation is difficult due to the variability of the resonator properties as a result of variations in the fabrication process and external conditions. The most effective excitation is achieved at the critical coupling condition when the intrinsic loss rate $\kappa_i$ of the resonator is matched to the external coupling rate $\kappa_{ex}$ to a bus waveguide[1–5]. We can thus realize critical coupling by controlling the intrinsic loss and/or the external coupling rate[6–8] to ensure optimal excitation. The intrinsic loss rate is primarily determined by material absorption and scattering loss due to geometry and surface roughness[9]. References 10,11 showed that it is possible to engineer the intrinsic loss rate to match the external coupling by smoothing relevant interfaces or adding semiconductor optical

amplifiers. Nevertheless, controlling these mechanisms with a high degree of accuracy in real-time, as required for high-quality factor (high-Q) resonators, is challenging. Another recent approach to attain critical coupling relies on the integration of 2D materials in photonic devices to control material absorption[5,8,12–14]. However, this process necessitates intricate fabrication procedures at the cost of reducing the quality factors. Numerous authors have shown that it is possible to tune the external coupling by designing the gap between the bus waveguide and the resonator[10,15–17]. This tuning is also challenging, since the coupling rate can be sensitive to the gap and to fabrication imperfections due to the field confinement[18,19]. In addition, once the photonic circuit is fabricated, the coupling rate cannot be easily adjusted. Therefore, uncertainties in the fabrication process make it

[1]Department of Electrical Engineering, Columbia University, New York, NY 10027, USA. [2]Photonics Initiative, Advanced Science Research Center, City University of New York, New York, NY 10031, USA. [3]Department of Applied Physics and Applied Mathematics, Columbia University, New York, NY 10027, USA. [4]Physics Program, Graduate Center, City University of New York, New York, NY 10016, USA. [5]These authors contributed equally: Jakob Hinney, Seunghwi Kim. ✉ e-mail: aalu@gc.cuny.edu; ml3745@columbia.edu

difficult to reliably achieve critical coupling in photonic devices. Within some geometries, it may be possible to control the ring coupling using microheaters to achieve critical coupling[20,21]. However, this approach introduces complexity into the design and fabrication processes, and requires external power.

An alternative route to efficient excitation and energy storage inside a resonator involves tailoring the excitation signal and shaping the incoming pulse in time, rather than changing the system geometry. This approach is based on virtual critical coupling[22], which is rooted in the scattering response of resonant systems driven by impinging waves oscillating at tailored complex frequencies. By tuning the complex frequency excitation, it is possible to induce an effect analogous to introducing either material gain or loss, enabling a generalized form of critical coupling that supports tunable and efficient resonant excitation without having to physically manipulate the system parameters. As the resonator reaches a quasi-steady state response, oscillating at the same complex frequency as the excitation signal, its intrinsic loss rate is controlled by the imaginary part of the complex frequency[23]. Recent studies have explored such control to realize coherent absorption[24,25], PT-symmetry[26], pulling optical forces[27], exotic scattering features[28], and loss compensation[29], without relying on material gain or loss but instead controlling effective gain and loss through the temporal excitation dynamics.

In this work, we theoretically and experimentally demonstrate the effective tuning of the resonator loss rate in an integrated photonics setting operating at around 1550 nm, through the precise control of the excitation signal. We achieve virtual critical coupling and hence enhance the efficiency with which we can couple light to nanoscale photonic resonators. Our technique further enables the manipulation of output pulses measured after passing through a resonator, showcasing nontrivial generation of time-reversal pairs. Importantly, the simplicity of our approach indicates that it does not require complicated settings, and it can be readily extended beyond integrated photonics.

## Results
### Virtual critical coupling

We study the device shown in Fig. 1a, consisting of a ring resonator with intrinsic loss rate $\kappa_i$ side-coupled to a waveguide at an external coupling rate $\kappa_{ex}$. Optical signals transiting through the bus waveguide can couple to the resonator and excite its modes. As mentioned above, the impinging energy is maximally coupled to a given mode if the input signal oscillates at its resonance frequency and the coupling rate is identical to the internal loss rate, such that $\kappa_i = \kappa_{ex}$, meeting the critical coupling condition. Away from critical coupling, a finite amount of impinging power does not couple to the resonator and is collected at the other end of the waveguide [see Fig. 1a]. When exciting at

resonance, we observe a nonzero transmission dip $T(\Delta = 0) = |\frac{\kappa_i - \kappa_{ex}}{\kappa_i + \kappa_{ex}}|^2$ for $\kappa_i \neq \kappa_{ex}$ under a monochromatic excitation, as shown in Fig. 1b, where $\Delta = \omega_0 - \omega_L$ is the laser detuning, and $\omega_0$, $\omega_L$ are the resonance and pump frequencies, respectively. In contrast, the transmission dip reaches zero at $\Delta = 0$ under the critical coupling condition $\kappa_i = \kappa_{ex}$ [black dashed curve in Fig. 1b]. Here we demonstrate that it is possible to tune the transmission dip to zero in a non-critically coupled resonator by shaping the excitation signal in time, such that the incoming light oscillates at a complex frequency whose imaginary part compensates for the mismatch between loss and coupling rates.

We first theoretically analyze the temporal response of the transmitted signal measured at the end of the bus waveguide for a general excitation signal oscillating at a complex frequency $e^{-i\omega_L t}$ where $\omega_L = \omega_{re} + i\omega_{im}$. The imaginary part $\omega_{im}$ indicates the growth (or decay when negative) rate of the signal amplitude modulation. Under suitable conditions on the resonator dispersion and on the chosen complex frequency of excitation, the transmission coefficient, defined as the ratio of measured field and input field at the same instant in time can be written as (see Supplementary Section 1 for details)

$$\tilde{T}(t) \equiv 1 - \frac{2\kappa_{ex}}{\kappa + i\Delta + \omega_{im}}[1 - e^{-(\kappa + i\Delta + \omega_{im})t}] = \tilde{T}_{ts}(t) + \tilde{T}_{qss}. \tag{1}$$

This quantity generally varies in time and it can be decomposed into two portions: the transient response $\tilde{T}_{ts}(t) = \frac{2\kappa_{ex}}{\kappa + i\Delta + \omega_{im}}e^{-(\kappa + i\Delta + \omega_{im})t}$, and the quasi-steady state response $\tilde{T}_{qss} = \frac{\kappa_i - \kappa_{ex} + i\Delta + \omega_{im}}{\kappa_i + \kappa_{ex} + i\Delta + \omega_{im}}$, which oscillates in time at the same complex frequency as the excitation. We define transmission as the squared magnitude of this coefficient, i.e., $T(t) = |\tilde{T}(t)|^2$. Further, we assume that the cavity has zero fields at $t = 0$, when the excitation starts, and that the excitation starts sharply at $t = 0$ as a complex frequency signal. Other assumptions on the initial conditions and the transition at $t = 0$ affect the transient component, but do not modify the quasi-steady state response $\tilde{T}_{qss}$. At resonance ($\Delta = 0$), the quasi-steady state can become identically zero if the imaginary part of the excitation frequency obeys the virtual critical coupling condition

$$\omega_{crit} = \kappa_{ex} - \kappa_i. \tag{2}$$

By adjusting the input signal time constant $\tau_{in} = 1/\omega_{im}$, which is the inverse of the imaginary part of the impinging complex frequency, we can achieve critical coupling for an overcoupled or undercoupled cavity when the input time constant is identical to the critical time constant: $\tau_{in} = \tau_{crit} = 1/\omega_{crit}$. Figure 1c shows the on-resonance transmission for a monochromatic (i.e., $\omega_{im} = 0$ and $\omega_{re} = \omega_o$) resonant excitation, starting at $t = 0$. The dashed curve represents critical

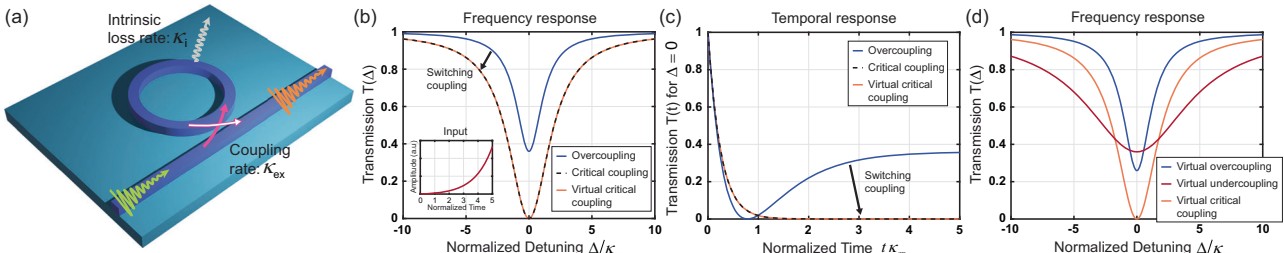

**Fig. 1 | Frequency and temporal responses of a microring resonator excited with complex frequency signals. a** Schematic of a microring resonator with intrinsic loss $\kappa_i$ side-coupled to a bus waveguide with coupling rate $\kappa_{ex}$. An impinging pulse is launched through the bus waveguide to excite the resonator, and the signal is measured at the end of the waveguide. **b** Transmission spectrum of an overcoupled resonator ($\kappa_{ex} > \kappa_i$, blue) and quasi-steady state transmission of the same resonator excited with a tailored complex frequency pulse (inset) leading to

virtual critical coupling (orange), compared to CW-transmission of a critically coupled resonator (dashed). **c** Temporal evolution of the transmission for overcoupled (blue), critically coupled (dashed), and virtually critically coupled (orange) scenarios excited at resonance ($\Delta = 0$). **d** Frequency spectrum of the transmission for virtually critically coupled (orange), virtually overcoupled (blue), and virtually undercoupled (red) scenarios. By varying the growth rate of the input signal, it is possible to control the coupling regime between resonator and waveguide.

coupling where $\kappa_{ex} = \kappa_i$ and, as expected, the transmission is unitary at the start and rapidly converges to zero after a transient period of a few multiples of the time constant $\tau$. In contrast, for the overcoupled scenario, for which $\kappa_{ex} > \kappa_i$ (blue curve), the transmission converges to a nonzero steady state $\lim_{t \to \infty} T(t) = |\frac{\kappa_i - \kappa_{ex}}{\kappa_i + \kappa_{ex}}|^2$, with a nonzero transmission dip in the frequency response in Fig. 1b. An undercoupled cavity ($\kappa_{ex} < \kappa_i$) would show a similar response, with detailed analysis discussed in Supplementary Section 3.

By exciting at a complex frequency, we can tune the system to meet the virtual critical coupling condition in Eq. (2), thereby exciting the overcoupled cavity more efficiently. The orange curve of Fig. 1c illustrates the temporal evolution of the squared magnitude of the transmission coefficient in Eq. (1) under this condition, and we indeed observe that its response is identical to the response of the critically coupled resonator [black dashed curve of Fig. 1c], showing how the transient transmission rapidly decays to a zero quasi-steady-state. This result is consistent with the frequency response of the overcoupled resonator calculated at this complex frequency of excitation, shown in Fig. 1b (see Supplementary Section 2 for detailed discussion). Generally, this result shows that an input pulse with growing time constant $\tau_{in} = \tau_{crit}$ can restore the conjugate matching condition by compensating the insufficient (or excessive) loss rate with virtual loss (or gain)[22,30]. In addition, by adiabatically varying in time the growth rate of impinging signals, we may dynamically control the effective intrinsic loss, varying the coupling regime. Figure 1d provides an illustrative example of this control, showing three transmission dispersions calculated for different values of $\tau_{in}$ for the same resonator, resulting in virtual critical coupling at $\tau_{in} = \tau_{crit}$ (orange), virtual overcoupling (blue) at $\tau_{in} = 10\,\tau_{crit}$, and virtual undercoupling (red) at $\tau_{in} = 0.2\,\tau_{crit}$. A more detailed discussion of these scenarios is presented in Supplementary Section 2.

## Experimental demonstration

We demonstrate virtual critical coupling in a silicon nitride (SiN) microring resonator coupled to a bus waveguide of the same cross-section (1500 nm wide by 730 nm tall) as shown in Fig. 2a. The SiN waveguides are clad with $SiO_2$ on a silicon substrate as in Fig. 2b.

Further fabrication and measurement details are provided in the Methods section. For all measurements we excite the fundamental transverse electric (TE) mode supported by the resonator, and a full-wave simulation of its modal profile is presented in Fig. 2b. We first characterize our cavity by inputting and rapidly (less than 0.1 ns) switching off a monochromatic signal, in order to extract the cavity lifetime from the decaying optical fields [Fig. 2c] using a fast oscilloscope with a 30 GHz bandwidth. On a separate low-bandwidth photodiode, we also record the cavity transmission during the entire sequence to extract the cavity spectrum for real-frequency excitations [Fig. 2d]. The transmission dip and cavity lifetime measurement fully characterize the optical response of the overcoupled cavity, providing the intrinsic loss rate and bus coupling rate. Given the measured lifetime of the cavity $\tau = 1/(\kappa_i + \kappa_{ex}) = 119.4$ ps and the measured transmission dip at zero detuning $T(\Delta = 0) = |\frac{\kappa_i - \kappa_{ex}}{\kappa_i + \kappa_{ex}}|^2 = 0.58$, we estimate the intrinsic loss and bus coupling rates to be $\kappa_i/2\pi = 0.16$ GHz and $\kappa_{ex}/2\pi = 1.18$ GHz, respectively, indicating that the cavity is strongly overcoupled. The resonance we selected was in the telecom C-band, at a wavelength of 1559 nm, so we can correspondingly estimate the loaded quality (Q) factor as around $Q = \omega\tau/2 \approx 72{,}000$ (significantly lower than the intrinsic Q factor of roughly $Q_i = \omega/2\kappa_i \approx 600{,}000$ since the resonator is deeply overcoupled). Given the parameters extracted above, we can analytically plot the inverse of the transmission $|1/T(f)|$ in the complex frequency plane (where $f$ is the frequency detuning from resonance), observing the progression of coupling regimes as sketched in Fig. 2e. At the pole of the plane, the transmission converges to zero (blue circle) and thus virtual critical coupling is achieved.

For the pulse measurements, the input to the bus waveguide is modulated in intensity through an electro-optic modulator (EOM) with a 30 GHz bandwidth, allowing for input intensity shaping. The output of the bus waveguide is fiber-coupled, amplified by an erbium-doped fiber amplifier (EDFA), and recorded on the fast oscilloscope [see Fig. 3a and the Methods section]. We generate various input pulse excitations, each with tailored time constant $\tau_{in}$, and sweep the laser frequency in discrete steps to record the output at each carrier frequency. In order to obtain the transmission coefficient as a function of

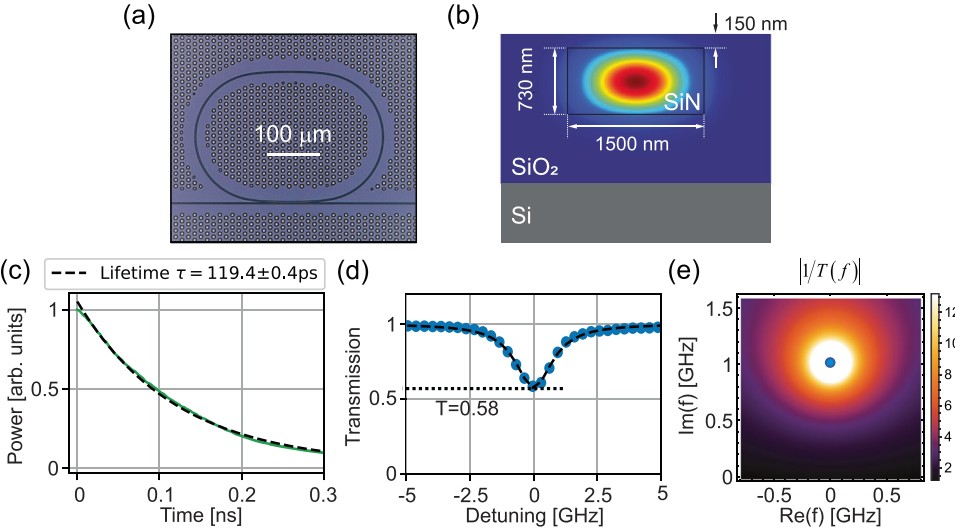

**Fig. 2 | Characterization of the microring resonator. a** Microscope image of the SiN integrated microresonator coupled to the bus waveguide. **b** Schematic of the cross-section for the 1500 nm wide × 730 nm high SiN microresonator covered with $SiO_2$, overlaid with the FDTD-simulated mode shape of the fundamental TE optical mode in the waveguide structure. **c** Cavity ring-down measurement, where the measured data (green line) is fitted (dashed line) to determine the lifetime of the cavity. **d** Measured transmission spectrum at real frequencies of the SiN resonator.

The intensity transmission dip at zero detuning ($\Delta = 0$) is 0.58. The estimated intrinsic loss and external coupling rates are $\kappa_i/2\pi = 0.16$ GHz and $\kappa_{ex}/2\pi = 1.18$ GHz; therefore, the resonator is strongly overcoupled to the bus waveguide. **e** Density plot of $|1/T(f)|$ in the complex frequency plane. The pole (brightest spot), which is marked by the blue circle, provides the condition for virtual critical coupling.

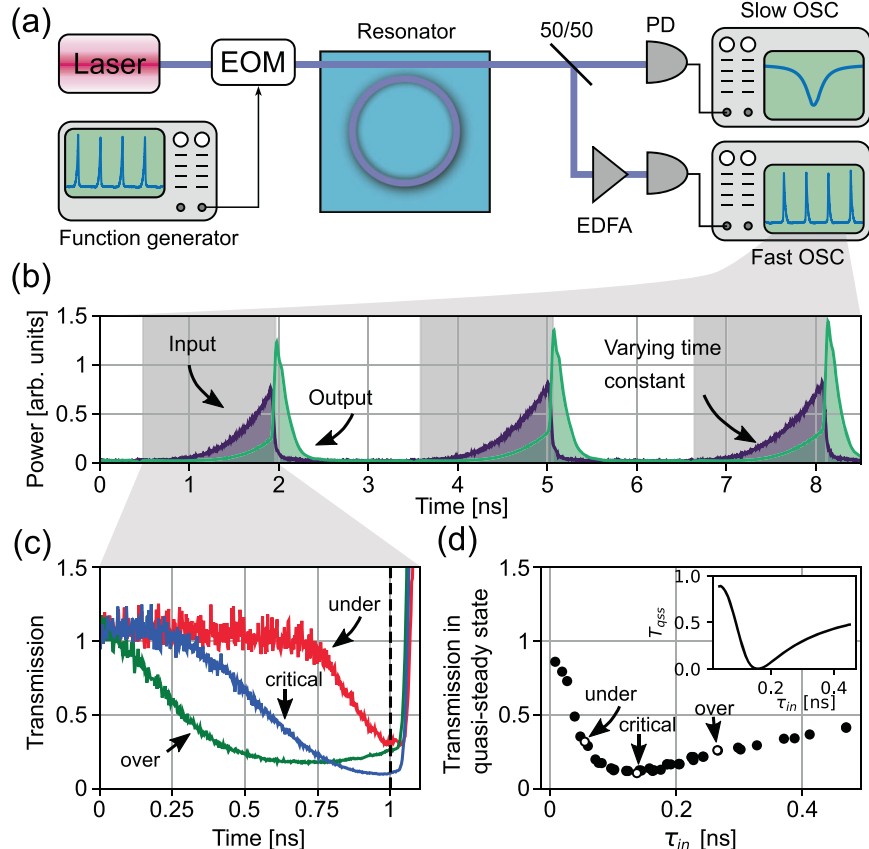

**Fig. 3 | Experimental setup and recorded transmission plots. a** Schematic of the experimental setup to control the complex frequency of excitation via an electro-optic modulator (EOM) and detect the transmitted signals with high temporal resolution through a fast oscilloscope (30 GHz), while monitoring the averaged transmission with a slow oscilloscope. PD photodetector, OSC oscilloscope, and EDFA erbium-doped fiber amplifier. **b** Sequence of input $I_{input}$ and output $I_{output}$ intensities for various pulse time constants. The energy stored in the resonator during the pulse decays partly into the bus waveguide and is observed as a sharp peak once the pulse ends. **c** Intensity transmission $T = I_{output}/I_{input}$ for virtual undercoupling (red, $\tau_{in} = 55ps$), virtual critical coupling (blue, $\tau_{in} = 138ps$), and virtual overcoupling (green, $\tau_{in} = 266ps$). **d** Measured intensity relative transmission in the quasi-steady state before switching off the pulse (vertical dashed line in Fig. 3c) as a function of $\tau_{in}$; the inset shows the theoretical value of the transmission in the quasi-steady state $T_{qss}$ for the same cavity parameters.

time during the pulses, we consider the ratio between input $I_{input}$ and output $I_{ouput}$ (measured at resonance) pulse intensities at the same time instant, both shown in Fig. 3b for three example pulses for illustration. The input signal $I_{input}$ is taken as the most off-resonance trace (corresponding to unity transmission in Fig. 2d) since there is minimal cavity interaction. The relative transmission $T = |\tilde{T}|^2 \equiv \frac{I_{output}}{I_{input}}$ is shown as a function of time in Fig. 3c for three selected example complex excitation frequencies. We extract the time constant of each input signal via fitting $I_{input}$ with exponential functions (in order to account for possible distortions from experimental limitations), and thus establish a connection between time constant $\tau_{in}$ and the relative intensity transmission $T$. Figure 3d then compares the relative transmission for different exponential time constants at the same point in time, corresponding to the dashed vertical line in Fig. 3c near the end of the pulses, which is sufficiently away from the start of the excitation to ensure that we have reached quasi-steady state. For comparison, the inset of Fig. 3d represents the theoretical curve for the transmission as a function of time constant $\tau_{in}$, in good agreement with our experimental data.

## Discussion

In Fig. 3c, we can clearly observe the effect of virtual critical coupling (blue curve). The quasi-steady state transmission coefficient is minimized at $T(\Delta = 0) = 0.11$ for an exponential pulse with a slope of $\tau_{in} = 138\,ps$, in good agreement with the theoretical value of $\tau_{crit} = 157\,ps$, calculated using the relation $\tau_{crit} = \tau/\sqrt{T(\Delta = 0)}$, derived

from Eq. (2). This effect demonstrates that the complex frequency excitation can compensate for the mismatch between bus coupling and intrinsic losses, and establishes virtual critical coupling, corresponding to vanishing transmission and maximum interaction between the waveguide and the resonator. The slight discrepancy between the predicted and measured transmission is likely caused by deviations in the exponential pulse shape. These translate into frequency broadening of the complex frequency component due to the bandwidth limits of the photodetector, oscilloscope, and EOM, and even nonzero group velocity dispersion (GVD) in the waveguide at the operating wavelength[31]. The pulse transmission at virtual critical coupling ($\tau_{in} = 138$ ps) flattens out around the minimum value, indicating that this transmission level is maintained as long as the exponential pulse growth and shape are maintained, which agrees with the theory. The steep rise in transmission towards the right of Fig. 3c corresponds to the end of the pulse, when the energy in the cavity is discharged into the bus. For time constants below and above $\tau_{in} = 138$ ps, the transmission reaches a local minimum but it then increases, as expected. These correspond to virtual undercoupling (shown here for $\tau_{in} = 55$ ps) and virtual overcoupling ($\tau_{in} = 266$ ps), respectively. For increasing $\tau_{in}$, the transmission increases asymptotically towards the transmission coefficient of 0.58 for real-frequency monochromatic excitations, seen in Fig. 2c and Fig. 3d, since larger time constants increasingly approximate a continuous-wave (CW) excitation. On the other hand, shorter time constants experience less extinction, as the pulse broadens spectrally and thus interacts less effectively with the cavity.

## Normalized stored energy $\eta$

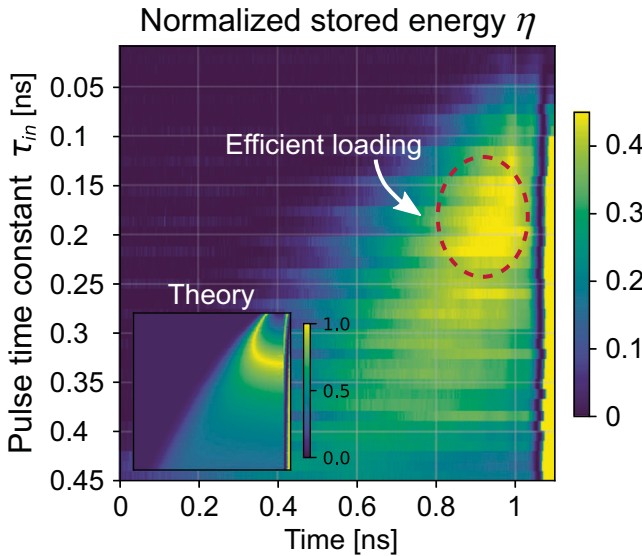

**Fig. 4 | Normalized stored energy $\eta$ in the resonator.** Effective cavity loading is enhanced in the marked area due to virtual critical coupling. Inset: theoretical calculation for the same parameter range.

In terms of our theory, too short of a pulse may not reach the quasi-steady state, in which case the response is dominated by the transient component in Eq. (1).

Additionally, we explore the field enhancement in the cavity at the virtual critical coupling condition. From the input and output of the resonator, we can infer the energy stored in the cavity using the dynamic equation $a_{out} = a_{in} - \sqrt{2\kappa_{ex}}a$. Here $a_{in}$, $a_{out}$, and $a$ correspond to the complex cavity input field, the bus output field, and the field amplitude in the resonator, respectively. The energy stored in the cavity is proportional to $|a|^2$, and we can define the normalized energy stored in the cavity as $\eta = \frac{2\kappa_{ex}|a|^2}{|a_{in}|^2} = |\sqrt{T} - 1|^2$. Figure 4 shows the measured normalized stored energy $\eta$ as a function of the input pulse time constant $\tau_{in}$ and time. Indeed, this quantity peaks around $\tau_{in} = \tau_{crit}$, as long as the system reaches quasi-steady state (towards the end of the pulses). The enhancement of energy storage is around 8 times larger than when the cavity is excited with a monochromatic CW excitation at the real-valued resonance frequency. Despite the bandwidth limitations of our measurement setup, we are able to observe significant enhancement of energy storage. The energy storage may be further enhanced using higher-Q devices since it requires longer-lasting growing pulses, significantly below the bandwidth limits set by the current setup. Hence, we can store almost the entire input energy as predicted in the theory (see the inset of Fig. 4), opening interesting opportunities in the context of efficiently engaging nonlinearities.

Complex frequency excitations also enable intriguing responses in the context of time-reversed waveforms to input pulses under virtual critical coupling. In the virtual critical case ($\tau_{in} = 138\,ps$), we expect from simulations to observe the flipped output of the input pulse in time, as sketched in Fig. 5 (a-left). In this illustration, the output starts decaying after the input pulse ends (at $t = 1\,ns$), and the rate of decay corresponds to the time constant of the cavity $\tau = 119.4\,ps$, confirmed by the analytical curve having the same decay rate (red dashed). The corresponding plot in Fig. 5b shows the experimental data, in agreement with the simulation. It is important to note that this time-reversal pair is not perfect, due to the intrinsic loss in the resonator, i.e., $\kappa_i \neq 0$. Interestingly, under the virtual critical condition, all impinging waves are fully trapped until the input is released at $t = 1\,ns$, which is not possible under a continuous-wave excitation in this system.

Interestingly, under different coupling conditions, which can be controlled through the excitation waveform, time-reversed outputs

can either compress or stretch temporally. For example, in the case of virtual overcoupling discussed above (at $\tau_{in} = 266\,ps$), the output waveform is compressed in time and its amplitude exceeds that of the input pulse as shown in Fig. 5 (a-middle). Conversely, for the virtual undercoupling at $\tau_{in} = 55\,ps$, the output pulse is temporally stretched, resulting in a broader waveform with reduced amplitude [see Fig. 5 (a-right)]. The decay rate for each of these is again $\tau = 119.4\,ps$, indicating the release of stored energy from the resonator into the waveguide with an eigenvalue corresponding to that of the resonator. Figure 5b again presents the corresponding experimental data for time-reversal pairs under the different virtual coupling conditions. Finite bandwidth and other experimental limitations cause slight distortions in the rising pulses, as well as nonzero switch-off times, but our measured data otherwise generally agree with the theoretical predictions.

In this work, we have demonstrated virtual critical coupling, undercoupling, and overcoupling by driving an overcoupled ring resonator with engineered pulses oscillating at complex frequencies. By tuning the temporal excitation, it is possible to tailor and compensate for the mismatch between resonator loss and coupling rate to a waveguide in an integrated photonics setup. Our experiments illustrate that it is possible to use this technique to maximize the stored energy inside a resonator, and dynamically tune its coupling to a waveguide without modifying its geometry. This drastic enhancement of intracavity fields can enable numerous opportunities in nonlinear optics and its broad applications, including optical switching[32] and optical quantum applications[33,34], even for cavities that are ordinarily too lossy or generally far from the critical coupling condition.

Virtual critical coupling fully confines light inside resonators without physically touching the systems. Specifically, it enables the trapping of light for the duration of impinging pulses, offering intriguing possibilities for optical delay lines and communication[35,36]. Importantly, our techniques pave the way for an array of applications related to real-time tuning within cavities. For instance, by exploring different impinging pulse scenarios, we can also achieve temporal compression or dilation of time-reversed waveforms, with potential uses including on-chip nontrivial tunable information processing without complicated structures[37,38]. The sensitivity of optical resonant sensors can also be enhanced through the tunability of bandwidths within on-chip integrated photonic circuits[39]. While the regimes of input intensities in this paper lie within the linear response of the resonators, virtual critical coupling allows for efficient storing of the input pulses inside the cavity, and for larger input intensities is expected to lead to efficient excitation of optical nonlinearities. The dynamics of such nonlinear resonances in the context of virtual critical coupling enable other exciting scenarios, such as in the context of pulse trapping and overcoming the delay-bandwidth limit[40–43].

While there has been significant research interest in complex frequency excitations, it has been argued that their realization can be particularly challenging in high-frequency regimes, especially in optics, compared to the acoustic or elastic wave domain[25,29]. Recent studies have explored alternative approaches to generate optical pulses with complex frequencies by summing over multiple real-frequency excitations in the mid-IR range. However, these methods require complicated post-processing of the measured signals, and a precise knowledge of the relative phase of the response at different real frequencies[39,44]. Our work demonstrates the ability to use complex frequency excitations in optics, applying them to the technologically-relevant near-1550 nm wavelength range and within on-chip integrated photonic circuits. Our techniques are facilitated by leveraging sophisticated photonic engineering of high-quality-factor microcavities and carefully tailored temporal excitations. In summary, our work introduces a versatile and powerful approach with far-reaching implications, ranging from improving light-matter interactions to enabling real-time adjustments within optical cavities for enhanced sensor performance and optical communication capabilities.

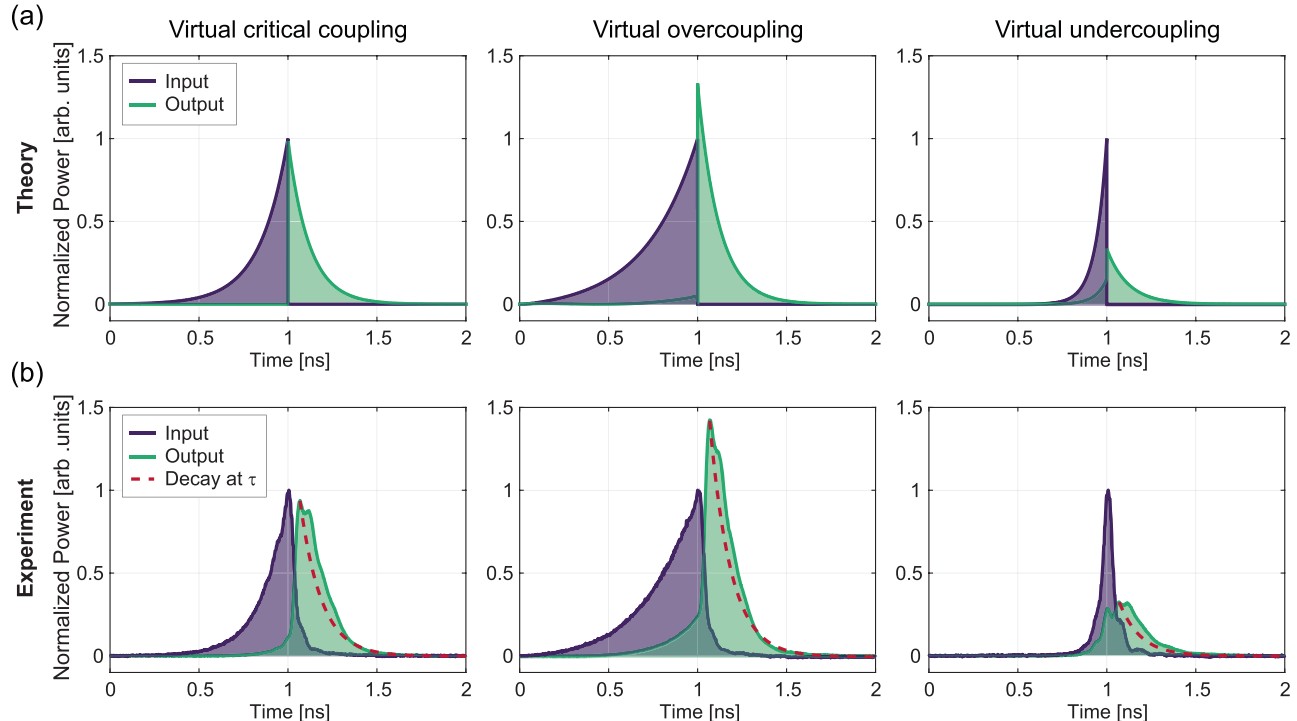

**Fig. 5 | Illustration of input and output pulses under various excitation scenarios.** Virtual critical coupling is demonstrated at $\tau_{in} = 138$ ps (left), virtual overcoupling is demonstrated at $\tau_{in} = 266$ ps (middle), and virtual undercoupling is demonstrated at $\tau_{in} = 55$ ps (right). **a** Theoretical predictions for the three cases. Inputs (purple), outputs (green), and analytical curves (red dashed) are represented. The analytical curves show that all outputs decay at the time constant of the cavity $\tau = 119$ ps. **b** Corresponding experimental results are displayed.

## Methods

### Device fabrication for SiN photonic circuit
To fabricate the devices, we began with low-pressure chemical vapor deposition (LPCVD) at 800 °C and annealing at 1200 °C in order to deposit 730 nm of silicon nitride (SiN) onto a thermal silicon dioxide layer. Using standard lithographic techniques, we patterned waveguides coupled to rings into the design shown in Fig. 2a, b and etched the SiN using reactive-ion etching into these shapes. We then deposited a layer of silicon dioxide on top (including on the sides of the waveguides) using plasma-enhanced chemical vapor deposition (PECVD) and planarized this top surface.

### Temporal measurement
To perform measurements of complex frequency excitations in SiN photonic circuits operating near 1550 nm, the experimental setup shown in Fig. 3a was used. We used a tunable diode laser (Toptica DLC CTL 1550) as the optical source. An arbitrary waveform generator (Keysight M8195A) was then used with an electro-optic modulator (EOM; EOSPACE 40+ Gb/s Intensity Modulator) to shape the laser output into pulses with the input signal time constant $\tau_{in}$. We used inverse waveguide tapers for both input and output coupling to the photonic circuit. A lensed fiber was used to couple to the on-chip input side of the SiN waveguide. The output light past the resonator was collected with a free-space lens and was evenly split and directed into slow and fast photodiode-oscilloscope pairs. The slow pair, consisting of a Keysight DSOX1204G oscilloscope and a Thorlabs PDA20CS2 photodiode, was used to monitor the steady-state transmission as the wavelength was swept through the resonance, in order to obtain a clean curve as in Fig. 2d. The fast pair, an optical module (80C02 with 30 GHz bandwidth) on a digital serial analyzer (Tektronix DSA8300), was concurrently used to record the time-dependent transmission during the actual sub-nanosecond pulses. This could have been used to

record the steady-state transmission as well but it was more convenient to use the combination of detectors. An erbium-doped fiber amplifier (EDFA; Amonics AEDFA-PA-35) was used to amplify the fast signal.

### Ring-down measurement for evaluating Q factor
A standard ring-down measurement was performed to determine the loaded Q factor of the resonator. We injected continuous input into the resonator and after switching this off with the EOM discussed above, we monitored the on-resonance transmission using the fast detector. The lifetime $\tau$ of the resonator was evaluated by fitting an exponential decay function to the cavity output. The loaded Q factor and the total loss were calculated using $Q = \omega\tau/2$. Here, the bandwidth of our EOM was 30 GHz, which corresponds to 33 ps, i.e., around one-quarter of the cavity lifetime. As a result, the EOM does not significantly affect the measurement.

## Data availability
All data that support the conclusions of this study in the paper and the supplementary materials are available from the corresponding author upon request.

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

## Acknowledgements

This work was supported by the Air Force Office of Scientific Research MURI program. M.L. acknowledges support from the U.S. Department of Energy, Office of Science, National Quantum Information Science Research Centers, Co-design Center for Quantum Advantage (C²QA) under contract number DE-SC0012704. Her work was also supported in part by the Center for Ubiquitous Connectivity (CUbiC), sponsored by the Semiconductor Research Corporation (SRC) and the Defense Advanced Research Projects Agency (DARPA) through the JUMP 2.0 Program. S.K. and A.A. acknowledge support from the Simons Foundation, a Vannevar Bush Faculty Fellowship, and the Air Force Office of Scientific Research.

## Author contributions

J.H., S.K., A.A., and M.L. conceived the idea. J.H and G.J.K.F. conducted the experiments and analyzed the data. S.K. performed the theoretical analysis. I.D. fabricated the devices. All authors contributed to writing the paper. M.L. and A.A. supervised the work.

## Competing interests

The authors declare no competing interests.
