## [Peer Review File · Nature Communications]

Efficient excitation and control of integrated photonic circuits with virtual critical couplingResponse to Referee's Comments on the paper "Efficient Excitation of Integrated Photonic Resonators with Virtual Critical Coupling"

Authors: J. Hinney, S. Kim, G. J. K. Flatt, I. Datta, A. Alù, M. Lipson

Reply to Referee A

Comment #1. *In the revised version of the manuscript 'Efficient Excitation of Integrated Photonic Resonators with Virtual Critical Coupling', the authors added new figure panels and the Methods section to improve the presentation. I appreciate the effort that authors have taken to improve the manuscript, both in content and clarity, however my concern remains about the practicality of this approach on account of the process occurring in a transient fashion. Also, even though the authors make a solid point about the significance of the demonstration, the prior publication of the idea still weighs significantly on novelty/impact – especially when considering the practical limitations of the method. Therefore, it is not possible to support the publication in Nature Photonics.*

Reply #1 We thank Referee A for acknowledging the effort invested in improving both the content and

clarity of our manuscript. While we understand and appreciate the concerns raised by the Reviewer on the novelty and practical limitations of our work, we respectfully disagree with their overall assessment.

As discussed in our previous rebuttal, it is crucial to emphasize that we have conducted a comprehensive examination of realistic scenarios for experimental implementation, an aspect that should not be understated, especially considering the inherent challenges associated with excitations at complex frequencies in practical scenarios when the carrier is at very high frequencies. Our exploration of various coupling regimes was not extensively addressed in the theoretical work Ref. [22] (which is a previous publication from our group). Furthermore, our experimental presentation of compressed or dilated time-reversal pairs in comparison to impinging waves introduces a perspective that has not been studied previously.

We are confident that our current work contributes novel perspectives in photonics and classical wave physics and, as such, we disagree with the comments by Referee A that our work lacks the requisite novelty and impact. We remain confident that our findings merit recognition for their innovative insights, theoretical advances and experimental demonstrations, and may raise significant attention within the broad physics community. We sincerely hope that Referee A may reconsider their stance, acknowledging that our paper aligns with the standards of Nature Communications in terms of both novelty and significance as we transfer our work to Nature Communications.

Reply to Referee B

Comment #1. *In their appeal, the authors provide a more detailed explanation of their experiments, answering several reviewers' concerns. I have appreciated the additional explanations in the response letter and the revised manuscript that enabled me to understand the work better. After reading the authors' arguments regarding the originality of the work, I am confident that the experimental verification of the published theoretical concept is worth publishing. The new results presented in Figure 5 further enhance this position.*

However, I was not convinced by the arguments presented regarding the importance and impact of the results. Many possible applications are listed in the abstract/introduction and the outlook, but none of them has been detailed in any way as to how it will be achieved. For example, the outlook section lists potential impacts in five different sentences. These include optical switching, quantum applications, optical delay lines, communications, on-chip information processing, pulse trapping, delay-bandwidth limit, light-matter interactions, and sensor performance. The big question is how exactly? How do the current results impact quantum applications? The response letter refers to two recent works, Science 381, 766 (2023) and arXiv:2307.09117 (2023); however, these works are based on surface-enhanced infrared absorption (SEIRA), which is not applicable in the telecom range. Ideally, I would like to see one crucial application being developed in more detail. Only in this way the reader can appreciate the power of the demonstrated concept.

Reply #1 We thank Referee B for finding that “the experimental verification of the published theoretical concept is worth publishing.” We appreciate that the new results in Figure 5 further support your position. We believe that our experimental demonstration represents a foundational contribution to the growing field of complex frequency excitations. In addition, our work offers a new perspective for opportunities aiming at enhancing the efficiency of state transfer in both quantum and classical systems, particularly operating near 1550 nm within integrated photonic settings, as discussed in our recent theoretical work [arXiv:2310.13454]. Our ability to trap specific wave packets of light through 'virtual' critical coupling in impedance-mismatched systems opens avenues for applications in slow light, enhancing the delay-bandwidth product, and more. In the realm of sensors, the manipulation and constriction of system bandwidth through complex frequency excitations, as illustrated in our manuscript (see Fig. S2 and Ref. [39]), offer opportunities for better sensitivities in resonant-based sensors.

As Referee B mentioned, the recent papers [Science 381, 766 (2023) and arXiv:2307.09117 (2023), now published in PRX 13, 041024 (2023)] experimentally operate at mid-IR and acoustic frequencies, and not in the telecom range. The mid-IR implementation, in addition, has to heavily rely on post-processing, as the authors were not able to excite the system with a complex frequency signal. We strongly believe that our current manuscript is pioneering in showcasing the potential applications of complex-frequency excitations at optical frequencies. Consequently, we feel that our work serves as the foundational step in illustrating the manifold possibilities emerging from this novel approach. As such, we feel that Nature Communications may be a good home for our paper, focusing on the foundations and physics advances more than the application aspects.

Comment #2. *Additional technical comments:*

- The authors should mention in the introduction the possibility of achieving critical coupling by dynamically tuning the resonators using microheaters, as has been widely used by some of the authors, e.g. Opt. Lett. 32, 3361 (2007).

- *Figure 5 shows three different excitation scenarios. However, the power is normalised. What is the normalisation factor? Is this factor the same across the three different cases?*

- *The new results in Figure 5 are presented as an example of time reversal phenomena. It is worth mentioning that this time reversal only happens for the exponential input that matches critical coupling and cannot be done with arbitrarily shaped pulses.*

Reply #2 We would like to thank Referee B for suggesting these improvements. We have now revamped the text by updating the references, including what the Referee suggested in Refs [20,21].

In Fig 5, we have normalized the curves to each maximum value of impinging waves, i.e., the impinging waves at $t=1$ ns. Thus, the normalization factors for the three cases are not the same.

Finally, we agree with Referee B that this time reversal only happens for pulse shapes matching critical coupling, as illustrated in Fig. 5, and the text reflects this important point.

Reply to Referee C

Comment #1. *The authors have gone far in addressing the comments and recommendations of the reviewers. As a result, the paper has improved considerably in clarity.*

Nevertheless the experimental data do not convince that the technique of virtual critical coupling has relevant applications. The arguments about relevance remain very abstract. The experimental results show that the quasi-steady-state of virtual critical coupling can only just be reached at the end of the pulse. It is left completely unclear whether there is any practical relevance for such a modality and whether a real quasi-steady-state is feasible for a prolonged period of time, as needed by a given application.

The results are obviously interesting and non-trivial from an experimental point of view but there is nothing particularly exciting about them since they confirm what basic theory about critical coupling and virtual critical coupling predicts.

Reply #1 We thank Referee C for finding our paper “interesting and non-trivial from an experimental point of view.” While we acknowledge Referee C's concerns regarding the potential limitations of practical implementations, particularly within short time ranges, we respectfully maintain our disagreement with the referee's perspective on the impact of our work.

We understand that maintaining a quasi-steady state for virtual critical coupling may be challenging, particularly as the pulses decay fast in time. However, this challenge can be overcome by employing higher Q cavity systems, for which the decay rate is slower. Within these systems, electro-optic modulators (EOMs) or other apparatus can generate pulses well before their exhaustion, allowing the quasi-steady state to be reached far before the end of pulses. We have thoroughly explained how systems can reach the quasi-steady state in both our main text and supplementary materials

More broadly, we believe that our current demonstration serves as a foundational step in illustrating a spectrum of potential applications arising from complex frequency excitations, as detailed in our response to Referee B. We are confident that our current work introduces novel perspectives to the fields of photonics and wave physics. Contrary to the comment by Referee C that there is nothing particularly exciting about them, we assert that our findings hold significant promise. We maintain our confidence that our work merits recognition for its innovative insights, theoretical advances and experimental demonstration. As such, we sincerely hope that Referee C may reconsider their stance, acknowledging that our paper aligns with the esteemed standards of Nature Communications in terms of both novelty and significance, as we progress with the transfer to Nature Communications.